# A Vortex-Assisted Dispersive Liquid-Liquid Microextraction Followed by UPLC-MS/MS for Simultaneous Determination of Pesticides and Aflatoxins in Herbal Tea

**DOI:** 10.3390/molecules24061029

**Published:** 2019-03-15

**Authors:** Rui Zhang, Zhen-Chao Tan, Ke-Cheng Huang, Yan Wen, Xiang-Ying Li, Jun-Long Zhao, Cheng-Lan Liu

**Affiliations:** 1Key Laboratory of Natural Pesticide and Chemical Biology, Ministry of Agriculture & Key Laboratory of Bio-Pesticide Innovation and Application of Guangdong Province, South China Agricultural University, Wushan Road 483, Guangzhou 510642, China; 13570447655@163.com (R.Z.); tanzhenchao@outlook.com (Z.-C.T.); 18131378639@163.com (Y.W.); lxykity521@163.com (X.-Y.L.); 17602038236@163.com (J.-L.Z.); 2Shenzhen Noposion Agrochemical Co. Ltd., Shenzhen 510640, Guangdong, China; kechenghuang@163.com

**Keywords:** vortex-assisted dispersive liquid-liquid microextraction, China herbal tea, pesticides residue, aflatoxins, UPLC-MS/MS

## Abstract

A method for detecting the organophosphorus pesticides residue and aflatoxins in China herbal tea has been developed by UPLC-MS/MS coupled with vortex-assisted dispersive liquid-liquid microextraction (DLLME). The extraction conditions for vortex-assisted DLLME extraction were optimized using single-factor experiments and response surface design. The optimum conditions for the experiment were the pH 5.1, 347 µL of chloroform (extraction solvent) and 1614 µL of acetonitrile (dispersive solvent). Under the optimum conditions, the targets were good linearity in the range of 0.1 µg/L–25 µg/L and the correlation coefficient above 0.9998. The mean recoveries of all analytes were in the ranged from 70.06%–115.65% with RSDs below 8.54%. The detection limits were in the range of 0.001 µg/L–0.01µg/L. The proposed method is a fast and effective sample preparation with good enrichment and extraction efficiency, which can simultaneously detect pesticides and aflatoxins in China herbal tea.

## 1. Introduction

Herbal tea is a kind of soup made with natural Chinese herbal medicine as raw materials according to the local climate, water and soil characteristics with the unique cooking methods by residents in the southern coastal areas of China [1,2,3,4], and guided by the theory of traditional Chinese medicine (TCM) health in the process of longer-term disease prevention and health care. It has the functions of clearing away heat and detoxifying, stimulating thirst, and preventing diseases [5,6,7] and as a social and recreational pastime [8,9,10,11]. It is also a widely used traditional health drink that has been widely circulated for generations.

Currently, there are many brands of Chinese herbal tea, mainly Wang Laoji, Jia Duo Bao, and Huang Zhenglong [12]. The herbal tea is composed of a variety of medicinal materials, which are susceptible to pests and diseases during growth and storage. It is necessary for using chemical pesticides during the cultivation of Chinese herbal medicines plants. Therefore, pesticides residues are inevitable in Chinese herbal materials [13,14,15]. Ultimately, the pesticide residues maybe also be detected in herbal tea. The organophosphorus pesticides (OPPs) were intensively applied at large-scale spraying on crops and sometimes they are detected in agricultural products. Although their residual time is short, OPPs can cause many acute and chronic neurotoxic diseases [16,17]. On the other hand, the herbal medicines plants are often infected some toxigenic fungi, such as *Aspergillus*, *Penicillium* and *Fusarium*. These fungi can produce mycotoxins during herbal plants growth and storage under suitable environmental conditions [18,19]. Aflatoxins (AFs) are secondary toxic metabolites mainly produced by *Aspergillus flavus* and *A. parasiticus*. The four main aflatoxins universally contaminated with food are AFB1, AFB2, AFG1, and AFG2, which were classified as class I human carcinogens by the International Agency for Research on Cancer in 1993 [20]. AFs not only harms the health of consumers but also causes the loss of economic benefits of Chinese herbal medicines. Hence, it is necessary to establish a rapid and effective method for detecting the OPPs and aflatoxins in the herbal tea.

Many studies have reported methods for the detection of trace pesticides, which some pesticides have endocrine activity (EDCs) [21] in Chinese herbal medicines: liquid chromatography (LC) and gas chromatography (GC) [22,23], gas chromatography-mass spectrometry (GC-MS) [24] and liquid chromatography-mass spectrometry (tandem) mass spectrometry (HPLC-MS or HPLC-MS/MS) [25]. The methods for detecting mycotoxins such as aflatoxins in traditional Chinese medicine have enzyme-linked immunosorbent assay (ELISA) [26], HPLC [27] and LC-MS/MS [28]. At the same time, pre-treatment procedure such as extraction and concentration are crucial to improving sensitivity and selectivity of the analytical methods owing to the presence of trace amounts of OPPs and aflatoxins and the complexity of real samples. The most frequently sample pre-treatment methods are solid phase extraction (SPE) [29] and liquid-liquid extraction (LLE) [30]. However, LLE has some disadvantage such as using a large volume of organic solvents and time-consuming. SPE needs to use expensive SPE cartridges. At present, a novel method named dispersive-liquid-liquid microextraction (DLLME) has been widely used to treat samples for pesticides residues, mycotoxins and plant ingredients analysis [31,32,33]. The DLLME have many advantages such as rapidity, simplicity, low cost, low solvent usage and high enrichment factor. However, most of the above-mentioned methods are only used for pesticides residues or mycotoxins [34,35,36,37].

Our study was to establish a vortex-assisted DLLME combined with UPLC-MS/MS method for detecting eight OPPs (dichlorvos, phoxim, Chlorpyrifos-methyl, chlorpyrifos, tolcofos-methyl, ediphenphos, ethion, and profenofos) and four aflatoxins (AFB1, AFB2, AFG1 and AFG2) in herbal tea. The important parameters of the DLLME procedure were optimized by single factor experiment and response surface design. The developed method was validated and applied to analyze the real herbal tea samples. To the best of our knowledge, it is the first report that a DLLME combined with UPLC-MS/MS method has been developed to simultaneously determine the pesticides residues and aflatoxins in herbal tea.

## 2. Materials and Methods

### 2.1. Chemicals and Standards

The dichlorvos (purity 98.0%), phoxim (97.0%), chlorpyrifos-methyl (99.7%), chlorpyrifos (99.8%), tolcofos-methyl (98.3%), ediphenphos (97.9%), ethion (99.1%), profenofos (99.0%), and aflatoxin B1 (99.0%), aflatoxin B2 (99.0%), aflatoxin G1 (99.0%) and aflatoxin G2 (99.0%) were bought from Dr. Ehrenstorfer (Augsburg, Germany). Acetonitrile (ACN, HPLC grade) and methanol (MeOH, HPLC grade) were purchased from Shanghai Anpel Scientific Instrument Corporation (Shanghai, China). Analytical-grade carbon tetrachloride (CCl_4_), chlorobenzene (C_6_H_5_Cl), chloroform (CHCl_3_) and dichloromethane (CH_2_Cl_2_), 1.1.2.2-tetrachloroethane (C_2_H_2_Cl_4_) were purchased from Tianjin Dongtian zheng Chemical Co. (Tianjin, China). Sodium chloride (NaCl) and hydrochloric acid (HCl) were purchased from Guangzhou Qian Hui Instrument Co., Ltd. (Guangzhou, China). Ultrapure water (UNIQUE-R20 purification system with UV + UF optional accessories, Research, Xiamen, China) was used in our work. A 0.22 mm cellulose membrane filter (Sterlitech, Kent, WA, USA) was used to filter the stock standard solution and herbal tea samples.

The stock solutions of eight target pesticides standards and four aflatoxins standards were prepared with acetone at 1000 mg/L and 100 mg/L, respectively, and stored in an amber glass vial at −20 °C. The working solutions were prepared by diluting the stock solution with acetonitrile.

### 2.2. Instruments and Equipment

The target analytes were determined with an ultra-performance liquid chromatography-tandem mass spectrometry (Waters TS-Q, Milford, MA, American). Separations were performed in an Acquity UPLC BEH C_18_ column (1.7 μm, 2.1 × 50 mm, Waters) under the condition of 40 °C. The mobile phase A was 2% (*v*/*v*) formic acid and the mobile phase B was methanol, at a flow rate of 0.3 mL/min. The injection volume was 5 μL. The elution solution was put into practice as follows: 0 min, 3% B; 0.5 min, 30% B; 6.5 min, 95% B; 7.5 min, 95% B; 9 min, 30% B and 10 min, 30% B.

The mass spectrometric analysis was carried out in the positive spray ionization mode and multiple reaction monitoring mode. Dry gas and atomizer are both nitrogen (N_2_). The optimal spray voltage was at 1.0 KV. Source and desolvation temperatures were 150 °C and 400 °C, respectively. The gas flow was at 650 L/h and the collision gas flow was at 0.25 mL/min. The achieved MS/MS parameters were generalized in Table 1.

### 2.3. Sample Preparation

Several herbal tea samples were collected from local supermarkets in Guangzhou, China. The samples were filtered using 0.22 mm cellulose membrane filters in order to remove some solid residues, and the filtered herbal tea samples were adjusted to pH 5.1 with 0.1 M of hydrochloric acid (HCl) or sodium hydroxide (NaOH).

### 2.4. Optimization of the Vortex Assisted DLLME Process

Chloroform (347 μL) (as extraction solvent) was added to 1614 μL acetonitrile (as dispersive solvent). The mixture was then injected into a 15 mL conical centrifuge tube that contained 5 mL herbal tea sample (pH 5.1). The tube was shaken for 60 s with a vortex mixer. A cloudy, turbid solution was rapidly obtained in the tube. Then the tube was centrifuged for 5 min at 3800 rpm. The upper aqueous phase was removed and the CHCl_3_ phase was quantitatively moved to a new centrifuge tube using a micro-syringe and evaporated to dryness under a stream of nitrogen at 45 °C. The evaporation residues reconstituted with 200 μL of acetonitrile. Finally, 5 μL was injected into the UPLC-MS/MS system for analysis.

### 2.5. Experimental Design and Data Analysis

In this study, the central composite design (CCD) was selected to optimize the three main factors that influenced the recovery efficiency (A: the sample pH; B: the volume of acetonitrile and C: the volume of CHCl_3_). The response value (Y) was the mean recoveries of twelve aimed compounds. According to the design, each of the three factors (A, B and C) was studied at five levels (Table 2). For each of the three studied variables, low and high set points were constructed for an orthogonal design (Table 2). The CCD design consisted of six replicates of the central points and twenty combinations. The resulting of twenty combinations, in which 5 mL of deionized water added into 0.01 mg/L of twelve analytes, were randomly performed. Every combination was done with three replicates and the obtained twelve analytes of mean recoveries were used as the response by statistical software. The relationship between the response and the three variables were expressed as the following quadratic polynomial equation:Y = b0 + b1A + b2B + b3C + b1b1A2 + b2b2B2 + b3b3C2 + b1b2AB + b1b3AC + b2b3BC(1)where Y is the response; A, B, C were the independent variables; b0 was the model intercept coefficient; b1, b2 and b3 were the linear coefficients; b1b1, b2b2 and b3b3 were the quadratic coefficients.

## 3. Results and Discussion

### 3.1. DLLME Procedure Optimization

The optimal DLLME conditions were determined for twelve targets in herbal tea samples, and the chloroform and acetonitrile be used as the extractant and dispersant, respectively. The influence of different parameters on extraction efficiency such as the extraction solvent types and volume, the dispersant types and volume, the aqueous phase pH were carefully investigated. In these experiments, a blank herbal tea sample spiked with 12 analytes (10 μg/L) was applied to assess the performance of the pre-treatment method and were calculated using the following equations:(2)$$\mathrm{EF}\;=\frac{C_{\mathrm{sed}}}{C_{0}}$$
(3)$$\mathrm{ER}\%=\frac{C_{\mathrm{sed}}V_{\mathrm{sed}}}{C_{0}V_{\mathrm{aq}}}\;\times 100\;=\mathrm{EF}\times \frac{V_{\mathrm{sed}}}{V_{\mathrm{aq}}}\times 100$$
where EF is the enrichment factor; C_sed_ is the concentration of target in the sedimentary phase; C_0_ is the initial concentration of target in the aqueous phase. ER% is the extraction recovery, V_sed_ is the volume of deposition phase, V_aq_ is the volume of the aqueous phase.

### 3.2. Extraction Solvent Selection

The extraction solvent plays a vital role in the extraction that affects the efficiency of microextraction specific condition. The appropriate extraction solvents must have a higher density than water, high partition coefficient, and poorly soluble in water. According to previous reports [38,39], we selected five halogenated hydrocarbons with a density higher than 1 g/mL, including carbon tetrachloride (CCl_4_), chloroform (CHCl_3_), dichloromethane (CH_2_Cl_2_), chlorobenzene (C_6_H_5_Cl) and 1.1.2.2-tetrachloroethane (C_2_H_2_Cl_4_). 5 mL of herbal tea (pH = 4, all the target analytes at 0.01 mg/L) was extracted by the mixture of 400 μL extraction solvent (carbon tetrachloride, chloroform, dichloromethane, chlorobenzene or 1.1.2.2-terachloroethane) and 800 μL of acetonitrile (dispersive solvent). The extraction efficiency was assessed by comparing the recovery rates of each compound. The results revealed that the extraction recoveries using CHCl_3_ as extraction solvent were higher than that of the other chlorinated solvent (shown as Figure 1a). The extraction recoveries of twelve compounds are 64.81%, 72.62%, 67.54%, 72.98%, 69.94%, 81.76%, 56.96%, 62.21%, 80.69%, 75.64%, 73.10% and 67.57%, respectively. Therefore, chloroform was selected as the extraction solvent.

### 3.3. Effect of the Extraction Solvent Volume (Chloroform)

The volume of extraction solvent is an important factor in the extraction recovery. Normally, a low volume of extraction solvent can achieve higher enrichment [40]. Increasing the volume of the extraction solvent would improve the extraction efficiency. However, they might decrease the enrichment factor [41]. In the second experimental step, we observed different volumes of CHCl_3_ (150, 200, 250, 300, 350, 400 and 450 μL) on the extraction efficiency at the same DLLME conditions. The results were shown in Figure 1b. The extraction recoveries for all analytes increased with increasing volume from 150 to 350 μL. However, the extraction recoveries of all analytes achieved a constant level under the volume above 350 μL. When the volume of chloroform was lower than 150 μL, it is difficult to withdraw the sedimentary phase. Thus, 350 μL was chosen as the optimal volume.

### 3.4. Effect of Dispersive Solvent

The dispersive solvent also obviously affect the extraction efficiency of DLLME. The role of the dispersive solvent converts the extraction solvent into droplets in an aqueous sample to help the analytes transfer into the organic phase. The dispersive solvents not only has a good solubility with the extraction solvent but also is miscible with water [42,43]. In the DLLME method, acetonitrile, acetone and methanol are usually chosen as the dispersive solvents. In our experiment, the effects of three dispersants (acetonitrile, acetone and methanol) on the extraction recovery were studied under the condition of 350 μL chloroform. The results showed that acetonitrile provided the highest extraction recoveries (Figure 1c). Thus, acetonitrile was chosen as the dispersive solvent.

### 3.5. Effect of Dispersive Solvent Volume (Acetonitrile)

The dispersive solvent volume directly affects the concentration of the extraction solvent in the aqueous phase, then affects the volume of deposition phase and extraction efficiency [44]. In order to get the optimal volume of acetonitrile, multiple experiments were carried out under different volumes of acetonitrile (600, 800, 1000, 1200, 1400, 1600, 1800, and 2000 μL) with 350 μL CHCl_3_. As results in Figure1d, the extraction recoveries increased with increasing volume of acetonitrile. However, the extraction recoveries decreased when the volume of acetonitrile was higher than 1600 μL. Therefore, 1600 μL was selected for the next study.

### 3.6. Effect of pH

The pH of the aqueous sample not only affects the presence of the target compounds (such as an ionic or neutral form) but also can change the distribution ratio of targets between the organic phase and the aqueous phase [45,46]. What is more, different herbal tea samples with different pH values may affect DLLME extraction efficiency. So, the effect of pH on the DLLME procedure was investigated by adding 0.1M HCl or NaOH into the herbal tea sample within the pH range of 2–8. As shown as Figure1e, the extraction recoveries for all analytes increased with the pH increased from 2 to 5. However, the extraction recoveries of all analytes presented a slowly declining trend with increasing pH (from 5 to 8). Furthermore, the standard error of each target was lower when the pH value was 5. Therefore, pH 5 was selected for the following experiments.

### 3.7. Experimental Design

The CCD was used to select the optimal experimental conditions and maximize recoveries. Three variables were studied by using the CCD at the center point at five levels with six replicates. The levels of each factor, high and low set points were established in orthogonal design (Table 2). The average recoveries of twelve targets were used as the parameters for the response surface curve, the polynomial regression analysis was performed on the response values in the experiment and the quadratic regression equation was obtained:R = 1.01 + 0.051A − 0.035B + 0.014C − 0.038AB + 0.019AC + 0.068BC − 0.13A^2^ − 0.90B^2^ − 0.12C^2^(4)where R is average recoveries of twelve analytes as a function of A (pH), B (volume of acetonitrile) and C (volume of CHCl_3_).

The method used ANOVA to evaluate the significance level and of each factor and interaction term, the larger the F value, the smaller the *p*-value, the more reliable the regression model obtained. As shown in Table 3, the model reached a very significant level with a *p*-value of less than 0.0001 and an F value of 77.74. The closer the decision coefficient (R^2^) of the model to 1, the closer the predicted value and the real values are 104.53% and 101.69%, respectively. The decision coefficient (R^2^) and the modified decision coefficient (AdjR^2^) of the model are 98.59% and 97.32%, respectively, and the coefficient of variation (CV) is 4.49%, that indicated a high correlation between the experimental and predicted values. The lack-fit-*p*-value of 0.0005 indicates that the model is susceptible to interference from non-experimental factor. The above test data showed that the model can reliably predict and analyze the mean recoveries of 12 targets by DLLME method. Figure 2 depicts the outline and 3D surface map of the mean recovery versus variable pair. The response surface map was applied to determine the extraction amount of eight pesticides and four mycotoxins on the interaction variables A–C. It could be seen from Figure 2a, the 3D map of the response surface model indicates that A (pH) and B (VE: volume of extraction solvent) have strong interactions. The target analytes get optimal extraction volume between 342 μL–350 μL and the most suitable pH range of 5.0–5.3. The A (pH) and C (VD: volume of dispersive solvent) have an impact on the mean recoveries of target extractants (shown in Figure 2b). When the B (VE: volume of extraction solvent) was fixed value, the extraction rate increased in the range of 1541 μL–1600 μL for acetonitrile. However, the extraction rate decreased at the volume of acetonitrile more than 1600 μL. Figure 2c shows that the maximum recoveries were achieved at the volume of chloroform (B) and acetonitrile in the range of 342 μL–350 μL and 1600 μL–1629 μL, respectively, when A (pH) was a fixed amount. The optimal conditions for this model are pH 5.1, 347 μL of extraction solvent and 1614 μL of dispersive solvent.

### 3.8. Method Evaluation

For evaluating the suitability of the method for simultaneous analyze the organophosphorous pesticides and aflatoxins in herbal tea, the linear range, the detection limits (LODs), the quantification limits (LOQs), precision and accuracy of the method were investigated under optimal conditions.

Since the herbal tea contains sugar, pigments and other impurities, which may affect the chromatographic signal of the targets and cause matrix effects. The magnitude of matrix effects can be assessed by comparing the matrix-matching calibration curve with the solvent standard curve. If the slope is 0~±20%, it indicates that the sample has a weak matrix effect; if the slope is between ±20%~±50%, there is a medium matrix effect; if the slope exceeds ±50%, which shows a strong matrix effect [46]. Therefore, the standard solutions were prepared by using a blank matrix extract without the target analytes to eliminate and compensate for the matrix effect. The matrix effect formula was evaluated as follows:(5)$$\mathrm{ME}\;(\%)\;=\left(\frac{slope\;of\;the\;calibration\;curve\;in\;the\;matrix}{\;slope\;of\;the\;calibration\;curve\;in\;the\;solvent}-1\right)\times 100$$

According to Table 3, there was a weak matrix effect for all the targets, indicated that the matrix does not significantly interfere with the chromatographic signals of the targets. Due to the weak matrix effect, seven different concentrations of pesticides and mycotoxins (0.1–25 μg/L) were applied in acetonitrile. The 12 targets had a good linearity in the linear range (0.1 μg/L–25 μg/L) and the correlation coefficient (R^2^) higher than 0.9980. (Table 4). The LODs were ranged from 0.001 µg/L–0.01 µg/L at the signal-to-noise ratio of 3 (S/N = 3). The LOQs were used as the lowest added concentration in herbal tea samples and the LOQs of the aimed pesticides and aflatoxins were 1 μg/L and 0.2 μg/L, respectively.

To validate the vortex-assisted DLLME method, four levels of concentrations of the added recoveries experiments (*n* = 5) were carried out on two different kinds of herbal tea samples (Wang Laoji and Jia Duo Bao). The results of the recoveries and precision were indicated in Table 5. From the results as shown in Table 5, the average recoveries of all targets were between 70.06% and 115.65%, and the *RSDs* were low at 8.54%, in agreement with the established performance criteria [47]. When the addition of 500, 100 and 10 μg/L in the blank sample of herbal tea, the response value exceeded the linear range, and the dilution was 500, 100 and 10 times before injection.

### 3.9. Application of the Developed DLLME Method to Real Herbal Tea Samples

The established method was used to detect the target analytes (eight organophosphorus pesticides and four aflatoxins) in 10 batches of herb tea samples (five of Wang Laoji herbal tea beverages, five of Jia Duo Bao herbal tea beverages), which were purchased from local supermarkets on 2018 in Guangzhou, China. We found that four aflatoxins were not detected in all herbal tea samples. Traces of phoxim and chlorpyrifos were detected in one batch with concentrations of 0.16 × 10^−3^ mg/kg and 0.10 × 10^−3^ mg/kg, respectively. However, the concentration of two pesticides did not exceed the maximum residue limit (MRL) of phoxim (0.01 mg/kg) and chlorpyrifos (0.02 mg/kg) in Chinese herbal medicines formulated by the European Union.

## 4. Conclusions

In our study, a novel vortex-assisted DLLME method combined with UPLC-MS/MS was developed for the detection of organophosphorus pesticides and aflatoxins in herbal tea beverages. The response surface method based on the central composite design was used to optimize the important parameters affecting the extraction recoveries, so as to determine the interaction between variables and obtain the optimal experimental combination. The optimized extraction conditions were pH 5.1, 347 μL of chloroform and 1614 μL of acetonitrile. Under the optimum conditions, the linear range was 0.1–25 μg/L for all targets and the correlation coefficient (R^2^) > 0.998. The LOD ranged from 0.001 μg/L–0.01 μg/L (S/N > 3) and the quantitative limits were 1 μg/L for pesticides and 0.2 μg/L for mycotoxins. The fortified recoveries were 70.06%–115.65%, and the relative standard deviation is low at 8.54% (*n* = 5). The developed method has the advantages of simple and rapid operation, high concentration factor and environment friendliness. The method can simultaneously analyze and detect organophosphorus pesticides and aflatoxins in different kinds of herbal tea and liquid samples. As far as we know, this is the first time that vortex-assisted DLLME combined with UPLC-MS/MS is used to simultaneously detect the pesticides and aflatoxins in herbal tea.

## Figures and Tables

**Figure 1 molecules-24-01029-f001:**
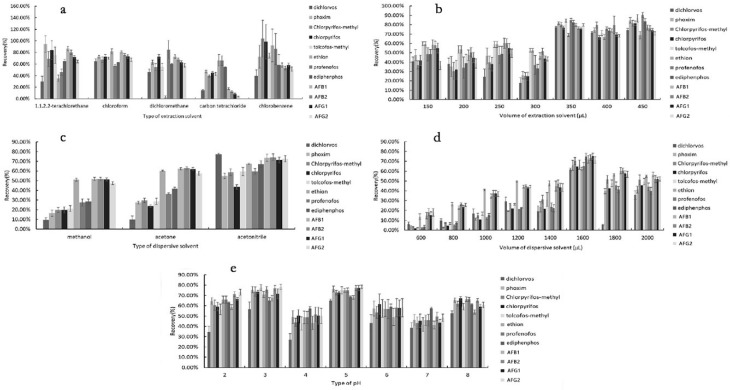
Optimized parameters of the DLLME procedure: (**a**) type of extraction solvent (**b**) volume of extraction solvent (μL) (**c**) type of dispersive solvent (**d**) volume of dispersive solvent (μL) and (**e**) pH. Extraction conditions: volume of chloroform, 350 μL; volume of acetonitrile,1600 μL; sample pH, 5; vortex-shaken time, 1 min; centrifuging for 5 min at 3800 rpm [a: sample pH 4, 400 μL of extraction solvent (carbon tetrachloride, chloroform, dichloromethane, chlorobenzene or 1.1.2.2-tetrachloroethane) and 800 μL of acetonitrile ; b: sample pH: 4, CHCl_3_ (150, 200, 250, 300, 350, 400 and 450 μL) and 800 μL of acetonitrile; c: sample pH: 4, 350 μL of chloroform and 800 μL of dispersive solvent (acetonitrile, acetone and methanol); d: sample pH: 4, 350 μL of chloroform and different volume of acetonitrile (600, 800, 1000, 1200, 1400, 1600, 1800 and 2000 μL); e: sample pH range of 2–8, 350 μL of chloroform and 1600 μL of acetonitrile].

**Figure 2 molecules-24-01029-f002:**
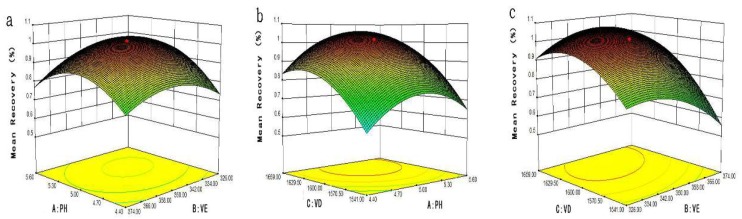
Response using the central composite design obtained by plotting: (**A**) pH; (**B**) VE: volume of extraction solvents; (**C**) VD: Volume of dispersive solvent, and (f) NaCl percentage vs. volume of dispersive solvent.

**Table 1 molecules-24-01029-t001:** The MS/MS parameters of the aimed pesticides and aflatoxins.

Analytes	Adduct On	Retention Time	Precursor Ion (m/z)	Product Ion (m/z)	Collision Energy/eV	Cone Voltage/V
dichlorvos	[M + H]^+^	2.89	221	79/109	34/22	30
phoxim	[M + H]^+^	5.41	299	129/153	13/7	30
*Chlorpyrifos-methyl*	[M + H]^+^	5.61	321.8	125/289.9	20/16	30
*chlorpyrifos*	[M + H]^+^	6.18	349.9	97/198	32/20	30
*tolcofos-methyl*	[M + H]^+^	5.47	263.9	79/109	36/22	30
ediphenphos	[M + H]^+^	5.24	311	109/111	32/26	30
*ethion*	[M + H]^+^	6.09	385	199.1/143	10/20	30
*profenofos*	[M + H]^+^	5.89	372.9	127.9/302.6	40/20	30
AFB1	[M + H]^+^	2.87	313.2	241.1/285.1	36/24	40
AFB2	[M + H]^+^	2.64	315.2	259.1/287.1	30/26	40
AFG1	[M + H]^+^	2.43	329.2	243.1/283.1	30/30	40
AFG2	[M + H]^+^	2.18	331.2	243.1/257.1	25/25	35

**Table 2 molecules-24-01029-t002:** The experimental range and levels of the variables in the central composite design (CCD).

variable	Parameter	Variable Levels				
		−α(low)	−1	0	1	+α(high)
A	pH	4	4.4	5	5.6	6
B	Volume of CHCI_3_ (μL)	310	326	350	374	400
C	Volume of ACN (μL)	1500	1541	1600	1659	1700

**Table 3 molecules-24-01029-t003:** Analysis of variance (ANOVA) for response surface quadratic model (12 analytes).

Source	Sum of Squares	d.f ^a^	Mean Square	F-Value ^b^	*p*-Value ^c^	Prof > F
Model	0.86	9	0.095	77.74	<0.0001	significant
A-pH	0.035	1	0.035	28.73	0.0003	
B-VE	0.016	1	0.016	13.33	0.0045	
C-VD	0.28	1	0.28	226.20	<0.0001	
AB	0.011	1	0.011	9.31	0.0122	
AC	3.306 × 10^−3^	1	3.306 × 10^−3^	2.48	0.14666	
BC	0.037	1	0.037	30.27	0.0003	
A^2^	0.25	1	0.25	203.26	<0.0001	
B^2^	0.12	1	0.12	94.77	<0.0001	
C^2^	0.20	1	0.20	164.98	<0.0001	
Redisual	0.012	10	1.205 × 10^−3^			
Lack of fit	0.012	5	2.393 × 10^−3^	38.76	0.0005	significant
Pure Error	3.082 × 10^−4^	5	6.165 × 10^−5^			
Cor Total	0.87	19				

^a^ Degrees of freedom. ^b^ Test for comparing model variance with residual (error) variance. ^c^ Probability of seeing the observed F-value if the null hypothesis is true.

**Table 4 molecules-24-01029-t004:** Calibration data of the DLLME procedure for pesticide and mycotoxins in Wang Lo Kat and Jia Duo Bao samples.

Samples	Analytes	Linearity (μg·L^−1^)	S(S_a_) ^a^	R^2^(Ra^2^) ^b^	Ratio (%)	Matrix Effect
Wang Laoji	dichlorvos	0.1–25	519,146(485,390)	0.9992(0.9994)	6.50	Mild
	phoxim	0.1–25	192,788(190,297)	0.9995(0.9995)	1.30	Mild
	*Chlorpyrifos-methyl*	0.1–25	89,333(85,254)	0.9996(0.9994)	4.57	Mild
	*chlorpyrifos*	0.1–25	183,791(172,781)	0.9997(0.9995)	5.99	Mild
	*tolcofos-methyl*	0.1–25	39,699 (38,327)	0.9990(0.9996)	3.46	Mild
	ediphenphos	0.1–25	750,665(761,053)	0.9993(0.9991)	1.38	Mild
	*profenofos*	0.1–25	160,427(157,929)	0.9990(0.9998)	3.72	Mild
	*ethion*	0.1–25	387,602(383,630)	0.9996(0.9995)	1.03	Mild
	AFB1	0.1–25	213,455(207,545)	0.9998(0.9998)	2.77	Mild
	AFB2	0.1–25	7042.5(7133.2)	0.9995(0.9995)	1.27	Mild
	AFG1	0.1–25	155,448(135,238)	0.9994(0.9996)	13.00	Mild
	AFG2	0.1–25	55,640(51,872)	0.9995(0.9995)	6.77	Mild
Jia Duo Bao	dichlorvos	0.1–25	519,146(470,095)	0.9992(0.9989)	9.45	Mild
	phoxim	0.1–25	192,788(185,570)	0.9995(0.9987)	3.74	Mild
	*Chlorpyrifos-methyl*	0.1–25	89,333(81,727)	0.9994(0.9989)	8.51	Mild
	*chlorpyrifos*	0.1–25	183,791(162,751)	0.9995(0.9991)	11.45	Mild
	*tolcofos-methyl*	0.1–25	39,699(36,143)	0.9996(0.9999)	8.96	Mild
	ediphenphos	0.1–25	750,665(760,249)	0.999(0.9991)	1.28	Mild
	profenofos	0.1–25	164,027(152,981)	0.9997(0.9998)	6.73	Mild
	ethion	0.1–25	387,602(370,279)	0.9996(0.9996)	4.35	Mild
	AFB1	0.1–25	213,455(189,305)	0.9998(0.9990)	11.31	Mild
	AFB2	0.1–25	7133.2(7457.9)	0.9995(0.9996)	4.55	Mild
	AFG1	0.1–25	155,448(136,653)	0.9994(0.9998)	12.09	Mild
	AFG2	0.1–25	55,640(50,823)	0.9995(0.9998)	8.66	Mild

^a^ S and R^2^, slope and determination coefficient of the calibration curves obtained from ACN solution. ^b^ S_a_ and R_a_^2^, slope and determination coefficient of the calibration curves obtained from matrix matched standard solutions.

**Table 5 molecules-24-01029-t005:** Recoveries of the OPPs and aflatoxins from herbal tea samples using optimized vortexed-assisted DLLME (*n* = 5).

Analytes	Spiked Level μg/L	Wang Laoji		Jia Duo Bao	
		Recovery (%)	RSD (%)	Recovery (%)	RSD (%)
dichlorvos	500	75.34	8.14	72.48	7.77
	100	70.12	5.57	71.61	6.83
	10	70.27	2.28	72.45	1.53
	1	70.06	2.95	70.44	3.86
phoxim	500	97.81	5.12	90.97	7.29
	100	79.78	5.33	74.50	4.85
	10	78.35	1.30	88.06	7.49
	1	71.19	4.43	70.21	4.54
*Chlorpyrifos-methyl*	500	82.39	5.17	89.05	8.26
	100	72.65	7.31	83.22	3.69
	10	77.15	3.14	79.53	7.45
	1	77.05	8.50	70.32	4.05
*chlorpyrifos*	500	84.84	3.00	83.11	5.92
	100	72.94	4.98	78.19	5.90
	10	74.63	2.39	86.64	5.03
	1	74.17	4.45	71.68	2.52
*Tolcofos-methyl*	500	75.70	8.28	104.34	5.03
	100	73.64	7.68	79.59	6.41
	10	76.23	5.92	76.23	5.92
	1	73.29	5.50	86.65	5.11
ediphenphos	500	71.04	3.63	72.28	2.51
	100	70.64	6.23	72.39	4.99
	10	71.80	6.96	70.44	4.69
	1	95.25	4.63	107.73	2.12
ethion	500	101.12	5.35	106.38	2.29
	100	78.08	4.68	82.26	4.05
	10	94.12	2.17	86.59	2.86
	1	115.65	2.40	114.09	2.21
profenofos	500	102.36	3.40	102.55	6.62
	100	81.89	5.28	78.78	3.29
	10	99.49	4.00	91.24	2.73
	1	81.88	5.29	83.59	4.72
AFB1	10	94.59	5.25	92.14	7.62
	1	83.28	3.83	77.44	2.22
	0.2	72.68	2.88	71.04	1.90
AFB2	10	101.19	5.22	97.07	5.12
	1	74.87	8.41	72.61	8.54
	0.2	73.56	4.76	75.81	3.42
AFG1	10	99.39	1.25	91.03	2.33
	1	70.67	4.07	70.67	2.32
	0.2	72.36	7.53	70.36	7.84
AFG2	10	102.99	2.62	92.15	2.46
	1	72.59	6.17	70.68	3.24
	0.2	70.10	2.70	72.40	1.89

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
