# Peer review of "A Vortex-Assisted Dispersive Liquid-Liquid Microextraction Followed by UPLC-MS/MS for Simultaneous Determination of Pesticides and Aflatoxins in Herbal Tea"

_molecules, 2019, doi:10.3390/molecules24061029_

Reviewer 1 Report

The paper is complete and all analytical parameters were reported and discussed.

In my opinion the paper can be accepted pending minor revisions, as below reported:

- for pesticides and, in general EDCs in environmental samples (line 57) add also the reference: Journal of Chromatography A, 1434, 1-18, 2016.

- for analytical procedures (line 64 after "selectivity" add the following references: 

Molecules, 23(9), article number 23092391, 1-3, 2018

Phytochemical Analysis, 29, 233-241, 2018

Microchemical Journal, 135, 33-38, 2017

Separations, 4(4), 36, 1-15, 2017

And consequently remove the more older references.

Author Response

Thank you very much for suggestions. We added two references (1. Journal of Chromatography A, 1434, 1-18, 2016; 2. Phytochemical Analysis, 29, 233-241, 2018).

Reviewer 2 Report

The manuscript by Zhang et al (molecules-395571) focuses on analysis pesticides and mycotoxins using DLLME technique followed by LCMS detection. Overall, the manuscript is well described.  However, the following minor changes are recommended before accepting the manuscript.

Line 74: UPLC/MS/MS format is not consistent with other places in the text. 

Line 116 and 135: Units I would either use “/” or “-1” to have all units in the same format. Please check this throughout the manuscript.  Also, no need a dot in between.

Line 112: Retention time or Recent time? Also, time units?

Explain why pH 5.1 works best.

Figure 1: font size is very small to read. I suggest making it a full-page figure.

Line 302: Table 5, RSD of Jia D. Buo 3.694% is not in the format as other values. Check this throughout the table. There are a few like that.

Line 314: HPLC or UPLC? Please confirm

Line 454: remove “…”

Author Response

Line 74: UPLC/MS/MS format is not consistent with other places in the text. Author’s response: Thank you very much for the helpful suggestions. We have changed the “UPLC/MS/MS” to “UPLC-MS/MS” and there all are consistent in the text. Line 116 and 135: Units I would either use “/” or “-1” to have all units in the same format. Please check this throughout the manuscript.  Also, no need a dot in between. Author’s response: We have changed the “0.1 mol·L-1 of hydrochloric” to “ 0.1 M of hydrochloric”. We used “/” as the same units in the text. We have checked this throughout the manuscript. Line 112: Retention time or Recent time? Also, time units? Author’s response: Thank you very much for the helpful suggestions. I am sorry that it is written mistaken. It is time units and we have changed the “Recent time” to “Retention time”. Explain why pH 5.1 works best. Author’s response: Thank you very much for the helpful suggestions. The single-factor experiments showed that the extraction recoveries of all analytes were the best when the pH was 5. Based the results of single-factor experiments, the experimental design have been done by the central composite design (CCD) to optimize the three main factors (pH, the volume of acetonitrile and CHCl3). The results showed the optimal pH conditions was 5.1. So, we select the pH 5.1 as the optimal pH conditions. Figure 1: font size is very small to read. I suggest making it a full-page figure. Author’s response: Thank you very much for the helpful suggestions. We changed it in revised manuscript/ Line 302: Table 5, RSD of Jia D. Buo 3.694% is not in the format as other values. Check this throughout the table. There are a few like that. Author’s response: Thank you very much for the helpful suggestions. We have checked all texts and unified the format in the revised manuscript. Line 314: HPLC or UPLC? Please confirm Author’s response: We used UPLC-MS/MS instrument in this study. We have changed “HPLC” to “UPLC” in the revised manuscript. Line 454: remove “…” Author’s response: I think that this sentence was added by Journal Editorial Department.